# Benzimidazole Bearing Thiosemicarbazone Derivatives Act as Potent α-Amylase and α-Glucosidase Inhibitors; Synthesis, Bioactivity Screening and Molecular Docking Study

**DOI:** 10.3390/molecules27206921

**Published:** 2022-10-15

**Authors:** Hayat Ullah, Shoaib Khan, Fazal Rahim, Muhammad Taha, Rashid Iqbal, Maliha Sarfraz, Syed Adnan Ali Shah, Muhammad Sajid, Mohamed F. Awad, Awatif Omran, Marzough Aziz Albalawi, Mahmoud A. Abdelaziz, Azza Al Areefy, Ibrahim Jafri

**Affiliations:** 1Department of Chemistry, University of Okara, Okara 56300, Pakistan; 2Department of Chemistry, Hazara University, Mansehra 21120, Pakistan; 3Department of Clinical Pharmacy, Institute for Research and Medical Consultations (IRMC), Imam Abdulrahman Bin Faisal University, Dammam 31441, Saudi Arabia; 4Department of Agronomy, Faculty of Agriculture and Environment, The Islamia University of Bahawalpur, Bahawalpur 63100, Pakistan; 5Department of Zoology, Wildlife and Fisheries, University of Agriculture Faisalabad, Sub-Campus Toba Tek Singh, Punjab 36050, Pakistan; 6Faculty of Pharmacy, Universiti Teknologi MARA Cawangan Selangor Kampus Puncak Alam, Bandar Puncak Alam, Selangor 42300, Malaysia; 7Atta-ur-Rahman Institute for Natural Product Discovery (AuRIns), Universiti Teknologi MARA Cawangan Selangor Kampus Puncak Alam, Bandar Puncak Alam, Selangor 42300, Malaysia; 8Department of Biochemistry, Hazara University, Mansehra 21120, Pakistan; 9Department of Biology, College of Science, Taif University, Taif 21944, Saudi Arabia; 10Department of Biochemistry, College of Science, University of Tabuk, Tabuk 71491, Saudi Arabia; 11Department of Chemistry, Alwajh College, University of Tabuk, Tabuk 71491, Saudi Arabia; 12Department of Chemistry, Faculty of Science, University of Tabuk, Tabuk 71491, Saudi Arabia; 13Clinical Nutrition Department, Applied Medical Science Collage, Jazan University, Jazan 45142, Saudi Arabia; 14Faculty of Home Economics, Nutrition & Food Science Department, Helwan University, P.O. Box 11795, Cairo 11281, Egypt; 15Department of Biotechnology, Faculty of Sciences, Taif University, Taif 21944, Saudi Arabia

**Keywords:** benzimidazole, thiosemicarbazone, synthesis, alpha-amylase, alpha-glucosidase, docking study, structure-activity relationship

## Abstract

Diabetes mellitus is one of the most chronic metabolic diseases. In the past few years, our research group has synthesized and evaluated libraries of heterocyclic analogs against α-glucosidase and α-amylase enzymes and found encouraging results. The current study comprises the evaluation of benzimidazole-bearing thiosemicarbazone as antidiabetic agents. A library of fifteen derivatives (**7**–**21**) was synthesized, characterized via different spectroscopic techniques such as HREI-MS, NMR, and screened against α-glucosidase and α-amylase enzymes. All derivatives exhibited excellent to good biological inhibitory potentials. Derivatives **19** (IC_50_ = 1.30 ± 0.20 µM and 1.20 ± 0.20 µM) and **20** (IC_50_ = 1.60 ± 0.20 µM and 1.10 ± 0.01 µM) were found to be the most potent among the series when compared with standard drug acarbose (IC_50_ = 11.29 ± 0.07 and 11.12 ± 0.15 µM, respectively). These derivatives may potentially serve as the lead candidates for the development of new therapeutic representatives. The structure–activity relationship was carried out for all molecules which are mainly based upon the pattern of substituent/s on phenyl rings. Moreover, in silico docking studies were carried out to investigate the active binding mode of selected derivatives with the target enzymes.

## 1. Introduction

Diabetes mellitus is a chronic endocrine disorder that affects the metabolism of carbohydrates, proteins, fat, electrolytes, and water. It includes a group of metabolic diseases characterized by hyperglycemia, in which blood sugar levels are elevated either because the pancreas does not produce enough insulin or because cells do not respond to the produced insulin [1]. Therefore, a therapeutic approach to treating diabetes is to decrease postprandial hyperglycemia [2]. This can be achieved by the inhibition of carbohydrate hydrolyzing enzymes like alpha-amylase and alpha-glucosidase [3]. Alpha-glucosidase and alpha-amylase are the important enzymes involved in the digestion of carbohydrates. Alpha-amylase is involved in the breakdown of long-chain carbohydrates, and alpha-glucosidase breaks down starch and disaccharides to glucose [4]. They serve as the major digestive enzymes and help in intestinal absorption. Alpha-amylase and alpha-glucosidase inhibitors are the potential targets in the development of lead compounds for the treatment of diabetes [5,6,7]. In diabetics, the short-term effect of these enzyme inhibitor drug therapies is to decrease high blood glucose levels [8]. The presently used synthetic enzyme inhibitors cause gastrointestinal side effects such as diarrhea, flatulence, abdominal bloating, etc. [9]. Subsequently, there is still a need to develop safer therapy.

Benzimidazole is a fused heterocyclic compound formed by the fusion of benzene and imidazole which shows a variety of biological activities such as anti-tumor [10], anti-glycation [11], antiprotozoal [12], anti-diabetic [13], anti-psychotic [14], anti-oxidant [15], and anti-viral [16]. Some anticancer medicines based on benzimidazole skeletons, such as Bendamustin [17], Hoechst 33258 [18], and Nocodazole [19], are in preclinical or clinical trials (Figure 1).

For many years, our research group has been working on the design and synthesis of heterocyclic analogues in search of potential lead molecules and has found promising results [20,21,22,23,24,25,26,27]. We have already reported some benzimidazole analogues as potent α-glucosidase, α-amylase [28], and α-glucosidase inhibitors [29] (Figure 2), but there is still a need to discover more derivatives for this inhibitory activity to classify lead candidates for more advanced research in the future. Thus, we decided to screen a library of benzimidazoles bearing thiosemicarbazone (**7**–**21**) for α-glucosidase and α-amylase inhibitory activities.

## 2. Results and Discussion

### 2.1. Chemistry

The series of benzimidazole-based thiosemicarbazone derivatives through a synthetic route was achieved in four steps. In the first step, 2-marcaptobenzimidazole (**1**) was treated with different substituted phenacyl bromide (**2**) (1 mmol) in ethanol in the presence of triethylamine and the mixture was refluxed for 3 h to give the first intermediate product (**3**) [30]. 

In the second step, hydrazine hydrate (**4**) and carbon disulfide mixed in ethanol under refluxed conditions for 3 h, afforded the second intermediate product (**5**) [31].

In the third step, intermediate products (**3**) and (**5**) were mixed in ethanol, followed by the addition of acetic acid under refluxed conditions for about 4 h to yield the third intermediate product (**6a**–**o**)**.** Finally, the intermediate product (**6a**–**o**) was treated with different substituted benzaldehydes in ethanol in the presence of acetic acid as a catalyst and refluxed the mixture for about 4 h to yield benzimidazole-bearing thiosemicarbazone derivatives (**7**–**21**) as a final product (Figure 1, Table 1). After the completion of reactions, the final products were then washed with *n*-hexane to get pure compounds. The primary confirmation of product was done with the help of TLC, which was a clear indication of single spot final product formation when compared with the precursor spots. However, further confirmation of the products was done through NMR spectroscopy.

The ^1^H NMR spectrum of derivative **17** was recorded in DMSO-*d*_6_ on a Bruker 600 MHz instrument. The peak for the benzimidazole N-proton (benzimidazole-NH) was observed at δ_H_ 12.48 (s, 1H, NH). The most downfield singlets of two NH protons present on both sides of the thiol group (C=S) were resonated at δ_H_ 10.17 and 9.57, respectively. In addition, a CH=N proton gives a singlet at δ_H_ 8.88 chemical shift values. The molecule comprises three aromatic rings such as benzimidazole, *para*-nitrobenzene, and *para*-bromobenzene rings. Among *para*-nitrobenzene protons, the doublet appeared at δ_H_ 8.43 for two protons H-3 & H-5 (Ar-H), while the other two protons of this ring, H-2 & H-6 (Ar-H) were resonated at δ_H_ 8.39 (d, *J* = 8.5 Hz, 2H, Ar-H) as doublets, respectively.

On the other hand, two chemically equivalent aromatic protons Ar-H (H-2 & H-6) of the *para*-bromobenzene ring resonated at δ_H_ 8.17 as a doublet. Another doublet was also observed for the remaining two protons Ar-H (H-3 & H-5) of the *para*-bromobenzne ring at δ_H_ 8.01 chemical shift values. However, H-4/H-7 (Ar-H) of the benzimidazole ring resonated at 7.41 (d, *J* = 7.5 Hz, 2H, Ar-H), as a doublet. Moreover, there were multiple recorded for H-5/H-6 (Ar-H) of the same benzimidazole protons at δ_H_ 7.15–7.09. Furthermore, a singlet was also observed for two active methylene protons –CH_2_- attached between the sulfur and aromatic rings at δ_H_ 3.60 (s, 2H, -S-CH_2_).

As for compound **17**, the ^13^C NMR signals δ_C-13_ at 191.3 were attributed to thionyl carbon (C=S), while two peaks were observed for & (C=N) (C-NO_2_) at δ_C-13_ 167.0 and 159.3, respectively. Both bridged and substitution-bearing carbons of benzimidazole resonated at δ_C-13_ 147.1 (C), 136.0 (C), and 134.1 (C), respectively. Besides that, the peaks corresponding to the remaining four carbons of benzimidazole appeared at 121.5 (CH), 121.1 (CH), 113.3 (CH), and 108.1 (CH), respectively. The peak at δ_C-13_ 37.8 (-CH_2_-) corresponds to active methylene groups attached to sulfur atoms.

The peaks corresponding to carbons of *para*-bromobenzene ring-bearing substitutions resonated at δ_C-13_ 133.7 (C) and 123.0 (C). The signals for the other eight unsubstituted carbons of both para-nitrobenzene and para-bromo benzene rings have appeared at δ_C-13_ 133.3 (CH), 131.1 (CH), 130.8 (CH), 130.1 (CH), 129.8 (CH), 129.5 (CH), 129.3 (CH), and 127.4 (CH), respectively. A peak at δ_C-13_ 134.0 (C) was recorded for *para*-nitro benzene carbon, which is attached to (C-C=N).

### 2.2. In Vitro α-Glucosidase Activity

All the synthesized derivatives of benzimidazole-bearing thiosemicarbazone (**7**–**21**) were evaluated against α-glucosidase. All the derivatives showed good to excellent inhibitory potentials with IC_50_ values ranging from 1.30 ± 0.20 to 22.10 ± 0.20 µM as compared to the standard drug acarbose (IC_50_ = 11.29 ± 0.07 µM). Derivative **19** (IC_50_ = 1.30 ± 0.20 µM) is the most potent among the series, having floro moieties on both the aryl rings ‘A’ and ‘B’. A limited structure–activity relationship was carried out, which mainly depends upon the nature, number, position, and electron donating/withdrawing effects of the substituent/s on the aryl ring ‘A’/‘B’.

If we compare derivative **7** (IC_50_ = 14.20 ± 0.10 µM) with **8** (IC_50_ = 17.20 ± 0.10 µM) and **9** (IC_50_ = 14.20 ± 0.20 µM). The entire three derivatives have the same phenyl group on aryl ring ‘A’ and the nitro group on aryl ring ‘B’. The difference in the potentials of these derivatives may be due to the different position of the nitro group on aryl ring ‘B’ whereas the position of the phenyl group on aryl ring ‘A’ is the same in the entire derivatives (see Figure 3).

By comparing derivative **10** (IC_50_ = 11.20 ± 0.10 µM) having a phenyl group at a para position on aryl ring ‘A’ and hydroxy group at a para position on aryl ring ‘B’ with derivatives **12** (IC_50_ = 2.20 ± 0.20 µM) having two chloro groups at meta/para position on aryl ring ‘A’ and hydroxy group at a para position on aryl ring ‘B’, **16** (IC_50_ = 21.10 ± 0.01 µM) having a bromo group at the para position on aryl ring ‘A’ and a hydroxy group at para position on aryl ring ‘B’. All the derivatives have the same hydroxy group on aryl ring ‘B’ but the nature of the substituent/s on aryl ring ‘A’ is different. The difference in the activity of these derivatives may be due to the different types of substituent/s attached to the aryl ring ‘A’. The activity profile of analog-6 in this regard was found to be much more potent in the presence of hydroxyl and chloro moieties, which are responsible for making strong hydrogen bonds, thus showing an excellent biological profile when compared with the standard drug acarbose (see Figure 4).

Derivative **19** (IC_50_ = 1.30 ± 0.20 µM) having the flouro group at the *meta* position on aryl ring ‘A’ and ‘B’ with derivative **20** (IC_50_ = 1.60 ± 0.20 µM) also having the floro group at the *meta* position on aryl ring ‘A’ and the nitro group at the *para*-position on aryl ring ‘B’. The small difference in the activity may be due to the different nature of the moiety on aryl ring ‘B’ (see Figure 5).

### 2.3. In Vitro α-Amylase Activity

All the synthesized derivatives were also evaluated against the α-amylase enzyme and showed excellent activity, having an IC_50_ value in the range of 1.10 ± 0.20 to 21.10 ± 0.01 µM as compared to the reference drug acarbose (IC_50_ = 11.12 ± 0.15 µM).

By comparing derivative **11** (IC_50_ = 3.10 ± 0.10 µM) with **12** (IC_50_ = 1.10 ± 0.20 µM), **13** (IC_50_ = 2.10 ± 0.10 µM), and **14** (IC_50_ = 6.10 ± 0.20 µM). The entire derivative has two chloro groups at the *meta* and *para* positions on the aryl ring ‘A’ but the substituent on aryl ring ‘B’ is different. In derivatives **11**, **13**, and **14**, there is a nitro group at different positions on aryl ring ‘B’, while in derivative **12**, there is a hydroxy group on aryl ring ‘B’. The difference in the potentials of these derivatives may be due to the different nature and position of the substituent on aryl ring ‘B’ (see Figure 6).

Similarly, by comparing derivative **15** (IC_50_ = 17.10 ± 0.01 µM) with derivative **17** (IC_50_ = 15.10 ± 0.20 µM) and **18** (IC_50_ = 13.20 ± 0.20 µM). The entire derivative has the same bromo group on aryl ring ‘A’ and the nitro group at a different position on aryl ring ‘B’. A little bit of difference in the potentials of these derivatives may be due to the different positions of the nitro group on aryl ring ‘B’ (see Figure 7).

It was concluded from the whole study that a little bit of difference in the potentials observed may be due to nature, number, position, and the electron donating/withdrawing effect of substitution/s on aryl ring ‘B’. The binding interactions for all derivatives were confirmed through a molecular docking study.

### 2.4. Docking Study

Molecular docking was performed to investigate the interaction between synthesized compounds and targeted enzymes, i.e., α-amylase and α-glucosidase. The crystallographic coordinates of both enzymes were retrieved from the Protein Data Bank (www.rcsb.org). In this study, the docking procedure revealed that selected analogs showed excellent potential when tested against the targeted enzymes. Among the tested series, most analogs possessing varied functional groups, such as flouro and nitro-substituted, displayed significant potential with a superposed surface complex. Different substituted ring structures were docked and their binding modalities were observed against selected enzymes. In this regard, flouro and nitro-substituted analogs (**19** and **20**) exhibited better potential against enzymes. Specifically, in the case of efficacious compounds **19** and **20**, the protein–ligand interaction (PLI) profile not only listed the best potential (in silico) but also demonstrated better potency in vitro study.

The protein–ligand interaction profile for analog **19** bearing two flouro groups at the *meta*-position on ring-A and ring-B, while analog **20** had one flouro group at the *meta*-position of aromatic ring-A and the nitro group at the *para*-position on aromatic ring-B, respectively, exhibited different interactive residues for alpha-glucosidase, as shown in Figure 8. The interactive residues for analogue **19** such as ASN-475(halogen), PHE-476 (π-π stacked), ILE-233 (π-R), LYS-506 (π-R), SER-505 (Halogen), ASP-232 (π-Anion), ASP-568 (Attractive charges), TRP-432(π-S), ASP-496(π-Anion), and ASP-568(HB) Similarly, analogue **20** for alpha-glucosidase the interactive residues are SER-497(HB), SER-505(HB), ILE-233(π-R), LYS-506(π-R), ASP-232(HB), ASP-232(Attractive charges), ASP-568(HB), MET-470(R), TRP-432(π-S), and ASP-469(π-Anion), as shown in Figure 8.

The most potent compounds, **19** and **20**, displayed remarkable profiles due to various PLI profiles. Both analogues have the floro moiety at the *meta*-position on aryl ring ‘A’ while floro and nitro moieties are on aryl ring ‘B’, respectively. Analogue **19** had significant interactive residue for alpha-amylases such as THR-163 (Halogen), HIS-305 (π-π Stacked), TYR-151 (π-π T-Shaped), ILE-235 (π-R), LEU-162 (R), HIS-201(π-S), TRP-58(HB), HIS-299 (HB), ASP-356(Halogen), TRP-59(π-π T-Shaped), ASP-197(Attractive charges), and ASP-300(Attractive charges), as shown in Figure 9. While with analogue **20** against alpha-amylase, the residues are GLN-63(HB), TRP-59(π-π T-Shaped), HIS-305(π-π T-Shaped), GLY-306(U-Donor atom), GLU-240(Halogen), HIS-201(HB), ASP-300(HB), GLU-233(π-Anion), ASP-197(π-Anion), and LEU-162 (π-R), as shown in Figure 9.

The only differences found in both compounds **19** and **20** are the attached substituents and PLI profile, in both cases, two floro groups are attached to the *meta*-position on both aromatic ring **19**, while in the case of analog **20**, the flouro group is attached to the *meta*-position of aryl ring ‘A’ and the nitro group attached is to the *para*-position of aryl ring ‘B’. The floro group increases the nucleophilic character of the ring, while nitro, being an electron-withdrawing moiety, decreases the nucleophilic character; therefore, weak interaction was found in the case of analog **20**, but the presence of the benzimidazole moiety had a strong interaction with the active site of the enzyme. In addition, the docking results of selected compounds were compared with acarbose as a reference drug, the binding energy was found to be a few folds better than a standard drug, and the interaction of heteroatoms in the synthesized moiety was more significant (see Table 2 and Table 3).

## 3. Experimental

### 3.1. General Information

All chemicals and solvents were purchased from Sigma Aldrich (St. Louis, MO, USA) with a purity of 97 to 99%. NMR spectra were recorded on a Bruker Ultra shield Plus NMR spectrometer, using DMSO as a solvent. The chemical shifts were measured relative to TMS as an internal standard. The high-resolution mass spectra (electron impact, 60 eV) were run on a Finnigan MAT-311A instrument (Bremen, Germany). For visualization of the chromatogram, a UV lamp (Schimazdu, Duisburg, Germany) of wavelength 254/365 was used.

### 3.2. General Procedure for the Synthesis of Benzimidazole Bearing Thiosemicarbazone Derivatives (***7**–**21***)

Benzimidazole-based thiosemicarbazone derivatives were synthesized in four steps. In the first step, 2-marcaptobenzimidazole (1 mmol) was treated with different substituted phenacyl bromide (1 mmol) in ethanol (10 mL) in the presence of triethylamine and the mixture was refluxed for 3 h to give the first intermediate product. The crude product was filtered, washed with *n*-hexane and recrystallized in ethanol to obtain the pure product [30]. 

In the second step, hydrazine (2 mmol) and carbon disulfide (1 mmol) mixed in ethanol (10 mL) under refluxed conditions for 3 h, afforded the second intermediate product [31]. The crude product was filtered, washed with *n*-hexane and recrystallized in ethanol to obtain the pure product.

In the third step, equimolar intermediate products formed in first step and second step were mixed in ethanol (10 mL) followed by the addition of few drops of acetic acid under refluxed conditions for about 4 h to yield the third intermediate product.

Finally, the third intermediate product was treated with equimolar different substituted benzaldehyde in ethanol (10 mL) in the presence of few drops of acetic acid as catalyst and refluxed the mixture for about 4 h to yield benzimidazole bearing thiosemicarbazone derivatives as a final product. For every step the progress of reaction was confirmed with the help of TLC. The crude product was filtered, washed with *n*-hexane and recrystallized in ethanol to obtain the pure product. 

### 3.3. Spectral Analysis

#### 3.3.1. (*E*)-*N*′-(2-((1*H*-Benzo[d]imidazol-2-yl)thio)-1-([1,1′-biphenyl]-4-yl)ethylidene)hydrazinecarbothiohydrazide (**6a**)

*Yelowish solid*, Yield: 78%; ^1^H NMR (600 MHz, DMSO-*d*_6_): δ 12.30 (s, 1H, NH), 11.35 (s, 1H, NH), 9.41 (s, 1H, NH), 7.80 (d, *J* = 7.5 Hz, 2H, Ar-H), 7.70 (d, *J* = 7.3 Hz, 2H, Ar-H), 7.65 (dd, *J* = 7.2, 2.3 Hz, 1H, Ar-H), 7.40 (t, *J* = 6.9 Hz, 1H, Ar-H), 7.36–7.30 (m, 2H, Ar-H), 7.29 (d, *J* = 7.4 Hz, 2H, Benzo-Imidazole-H), 7.15 (d, *J* = 7.7 Hz, 2H, Benzo-imidazole-H), 3.50 (s, 2H, CH_2_), 3.10 (s, 2H, NH_2_), ^13^C NMR (150 MHz, DMSO-*d*_6_): δ 183.6, 154.7, 146.9, 142.5, 140.3, 138.6, 138.6, 132.2, 129.4, 129.4, 127.8, 127.8, 126.1, 126.1, 125.9, 122.3, 114.7, 114.7, 36.2. HR EIMS: *m*/*z* calcd for C_22_H_20_N_6_S_2_ [M]^+^ 432.0946.1241; Found: 432.0820.

#### 3.3.2. (*E*)-*N*′-(2-((1*H*-Benzo[d]imidazol-2-yl)thio)-1-([1,1′-biphenyl]-4-yl)ethylidene)hydrazinecarbothiohydrazide (**6b**)

*Yelowish solid*, Yield: 74%; ^1^H NMR (600 MHz, DMSO-*d*_6_): δ 12.31 (s, 1H, NH), 11.34 (s, 1H, NH), 9.41 (s, 1H, NH), 7.80 (d, *J* = 7.5 Hz, 2H, Ar-H), 7.70 (d, *J* = 7.3 Hz, 2H, Ar-H), 7.65 (dd, *J* = 7.2, 2.3 Hz, 1H, Ar-H), 7.40 (t, *J* = 6.9 Hz, 1H, Ar-H), 7.36–7.30 (m, 2H, Ar-H), 7.29 (d, *J* = 7.4 Hz, 2H, Benzo-Imidazole-H), 7.15 (d, *J* = 7.7 Hz, 2H, Benzo-imidazole-H), 3.50 (s, 2H, CH_2_), 3.10 (s, 2H, NH_2_), ^13^C NMR (150 MHz, DMSO-*d*_6_): δ 183.6, 154.7, 146.9, 142.5, 140.3, 138.6, 138.6, 132.2, 129.4, 129.4, 127.8, 127.8, 126.1, 126.1, 125.9, 122.3, 114.7, 114.7, 36.2. HR EIMS: *m*/*z* calcd for C_22_H_20_N_6_S_2_ [M]^+^ 432.0946.1241; Found: 432.0820.

#### 3.3.3. (*E*)-*N*′-(2-((1*H*-Benzo[d]imidazol-2-yl)thio)-1-([1,1′-biphenyl]-4-yl)ethylidene)hydrazinecarbothiohydrazide (**6c**)

*Brownish yellow solid*, Yield: 82%; ^1^H NMR (600 MHz, DMSO-*d*_6_): δ 12.34 (s, 1H, NH), 11.30 (s, 1H, NH), 9.41 (s, 1H, NH), 7.80 (d, *J* = 7.5 Hz, 2H, Ar-H), 7.70 (d, *J* = 7.3 Hz, 2H, Ar-H), 7.65 (dd, *J* = 7.2, 2.3 Hz, 1H, Ar-H), 7.40 (t, *J* = 6.9 Hz, 1H, Ar-H), 7.36–7.30 (m, 2H, Ar-H), 7.29 (d, *J* = 7.4 Hz, 2H, Benzo-Imidazole-H), 7.15 (d, *J* = 7.7 Hz, 2H, Benzo-imidazole-H), 3.50 (s, 2H, CH_2_), 3.10 (s, 2H, NH_2_), ^13^C NMR (150 MHz, DMSO-*d*_6_): δ 183.6, 154.7, 146.9, 142.5, 140.3, 138.6, 138.6, 132.2, 129.4, 129.4, 127.8, 127.8, 126.1, 126.1, 125.9, 122.3, 114.7, 114.7, 36.2. HR EIMS: *m*/*z* calcd for C_22_H_20_N_6_S_2_ [M]^+^ 432.0946.1241; Found: 432.0820.

#### 3.3.4. (*E*)-*N*′-(2-((1*H*-Benzo[d]imidazol-2-yl)thio)-1-([1,1′-biphenyl]-4-yl)ethylidene)hydrazinecarbothiohydrazide (**6d**)

*Yelowish solid*, Yield: 77%; ^1^H NMR (600 MHz, DMSO-*d*_6_): δ 12.38 (s, 1H, NH), 11.32 (s, 1H, NH), 9.41 (s, 1H, NH), 7.80 (d, *J* = 7.5 Hz, 2H, Ar-H), 7.70 (d, *J* = 7.3 Hz, 2H, Ar-H), 7.65 (dd, *J* = 7.2, 2.3 Hz, 1H, Ar-H), 7.40 (t, *J* = 6.9 Hz, 1H, Ar-H), 7.36–7.30 (m, 2H, Ar-H), 7.29 (d, *J* = 7.4 Hz, 2H, Benzo-Imidazole-H), 7.15 (d, *J* = 7.7 Hz, 2H, Benzo-imidazole-H), 3.50 (s, 2H, CH_2_), 3.10 (s, 2H, NH_2_), ^13^C NMR (150 MHz, DMSO-*d*_6_): δ 183.6, 154.7, 146.9, 142.5, 140.3, 138.6, 138.6, 132.2, 129.4, 129.4, 127.8, 127.8, 126.1, 126.1, 125.9, 122.3, 114.7, 114.7, 36.2. HR EIMS: *m*/*z* calcd for C_22_H_20_N_6_S_2_ [M]^+^ 432.0946.1241; Found: 432.0820.

#### 3.3.5. (*E*)-*N*′-(2-((1*H*-Benzo[d]imidazol-2-yl)thio)-1-(3,4-dichlorophenyl)ethylidene)hydrazinecarbothiohydrazide (**6e**)

*Light Yelow solid*, Yield: 73%; ^1^H NMR (600 MHz, DMSO-*d*_6_): δ 12.32 (s, 1H, NH), 11.42 (s, 1H, NH), 9.49 (s, 1H, NH), 7.82 (s, 1H, Ar-H), 7.79 (d, *J* = 7.1 Hz, 1H, Ar-H), 7.70 (d, *J* = 7.7 Hz, 1H, Ar-H), 7.39 (d, *J* = 7.3 Hz, 2H, Benzo-Imidazole-H), 7.10 (d, *J* = 7.5 Hz, 2H, Benzo-imidazole-H), 3.52 (s, 2H, CH_2_), 3.13 (s, 2H, NH_2_), ^13^C NMR (150 MHz, DMSO-*d*_6_): δ 183.9, 154.7, 146.7, 138.6, 138.6, 135.3, 133.1, 133.1, 130.3, 130.0, 126.2, 122.8, 122.8, 114.6, 114.6, 36.7. HR EIMS: *m*/*z* calcd for C_16_H_14_Cl_2_N_6_S_2_ [M]^+^ 424.1580; Found: 424.1490.

#### 3.3.6. (*E*)-*N*′-(2-((1*H*-Benzo[d]imidazol-2-yl)thio)-1-(3,4-dichlorophenyl)ethylidene)hydrazinecarbothiohydrazide (**6f**)

*Brownish yellow solid*, Yield: 71%;^1^H NMR (600 MHz, DMSO-*d*_6_): δ 12.36 (s, 1H, NH), 11.42 (s, 1H, NH), 9.49 (s, 1H, NH), 7.82 (s, 1H, Ar-H), 7.79 (d, *J* = 7.1 Hz, 1H, Ar-H), 7.70 (d, *J* = 7.7 Hz, 1H, Ar-H), 7.39 (d, *J* = 7.3 Hz, 2H, Benzo-Imidazole-H), 7.10 (d, *J* = 7.5 Hz, 2H, Benzo-imidazole-H), 3.52 (s, 2H, CH_2_), 3.13 (s, 2H, NH_2_), ^13^C NMR (150 MHz, DMSO-*d*_6_): δ 183.9, 154.7, 146.7, 138.6, 138.6, 135.3, 133.1, 133.1, 130.3, 130.0, 126.2, 122.8, 122.8, 114.6, 114.6, 36.7. HR EIMS: *m*/*z* calcd for C_16_H_14_Cl_2_N_6_S_2_ [M]^+^ 424.1580; Found: 424.1490.

#### 3.3.7. (*E*)-*N*′-(2-((1*H*-Benzo[d]imidazol-2-yl)thio)-1-(3,4-dichlorophenyl)ethylidene)hydrazinecarbothiohydrazide (**6g**)

*Yelowish solid*, Yield: 79%; ^1^H NMR (600 MHz, DMSO-*d*_6_): δ 12.39 (s, 1H, NH), 11.42 (s, 1H, NH), 9.49 (s, 1H, NH), 7.82 (s, 1H, Ar-H), 7.79 (d, *J* = 7.1 Hz, 1H, Ar-H), 7.70 (d, *J* = 7.7 Hz, 1H, Ar-H), 7.39 (d, *J* = 7.3 Hz, 2H, Benzo-Imidazole-H), 7.10 (d, *J* = 7.5 Hz, 2H, Benzo-imidazole-H), 3.52 (s, 2H, CH_2_), 3.13 (s, 2H, NH_2_), ^13^C NMR (150 MHz, DMSO-*d*_6_): δ 183.9, 154.7, 146.7, 138.6, 138.6, 135.3, 133.1, 133.1, 130.3, 130.0, 126.2, 122.8, 122.8, 114.6, 114.6, 36.7. HR EIMS: *m*/*z* calcd for C_16_H_14_Cl_2_N_6_S_2_ [M]^+^ 424.1580; Found: 424.1490.

#### 3.3.8. (*E*)-*N*′-(2-((1*H*-Benzo[d]imidazol-2-yl)thio)-1-(3,4-dichlorophenyl)ethylidene)hydrazinecarbothiohydrazide (**6h**)

*Lemon Yelow solid*, Yield: 76%; ^1^H NMR (600 MHz, DMSO-*d*_6_): δ 12.29 (s, 1H, NH), 11.42 (s, 1H, NH), 9.49 (s, 1H, NH), 7.82 (s, 1H, Ar-H), 7.79 (d, *J* = 7.1 Hz, 1H, Ar-H), 7.70 (d, *J* = 7.7 Hz, 1H, Ar-H), 7.39 (d, *J* = 7.3 Hz, 2H, Benzo-Imidazole-H), 7.10 (d, *J* = 7.5 Hz, 2H, Benzo-imidazole-H), 3.52 (s, 2H, CH_2_), 3.13 (s, 2H, NH_2_), ^13^C NMR (150 MHz, DMSO-*d*_6_): δ 183.9, 154.7, 146.7, 138.6, 138.6, 135.3, 133.1, 133.1, 130.3, 130.0, 126.2, 122.8, 122.8, 114.6, 114.6, 36.7. HR EIMS: *m*/*z* calcd for C_16_H_14_Cl_2_N_6_S_2_ [M]^+^ 424.1580; Found: 424.1490.

#### 3.3.9. (*E*)-*N*′-(2-((1*H*-Benzo[d]imidazol-2-yl)thio)-1-(4-bromophenyl)ethylidene)hydrazinecarbothiohydrazide (**6i**)

*Yelowish solid*, Yield: 73%; ^1^H NMR (600 MHz, DMSO-*d*_6_): δ 12.37 (s, 1H, NH), 11.35 (s, 1H, NH), 9.43 (s, 1H, NH), 7.65 (d, *J* = 7.3 Hz, 2H, Ar-H), 7.57 (d, *J* = 7.4 Hz, 2H, Ar-H), 7.36 (d, *J* = 7.3 Hz, 2H, Benzo-Imidazole-H), 7.08 (d, *J* = 7.5 Hz, 2H, Benzo-imidazole-H), 3.47 (s, 2H, CH_2_), 3.11 (s, 2H, NH_2_), ^13^C NMR (150 MHz, DMSO-*d*_6_): δ 183.7, 154.5, 146.4, 138.2, 138.2, 132.3, 131.3, 131.3, 128.1, 128.1, 125.2, 122.5, 122.5, 114.7, 114.7, 36.3. HR EIMS: *m*/*z* calcd for C_16_H_15_BrN_6_S_2_ [M]^+^ 433.9160; Found: 433.9080.

#### 3.3.10. (*E*)-*N*′-(2-((1*H*-Benzo[d]imidazol-2-yl)thio)-1-(4-bromophenyl)ethylidene)hydrazinecarbothiohydrazide (**6j**)

*Brownish Yelow solid*, Yield: 72%; ^1^H NMR (600 MHz, DMSO-*d*_6_): δ 12.31 (s, 1H, NH), 11.35 (s, 1H, NH), 9.43 (s, 1H, NH), 7.65 (d, *J* = 7.3 Hz, 2H, Ar-H), 7.57 (d, *J* = 7.4 Hz, 2H, Ar-H), 7.36 (d, *J* = 7.3 Hz, 2H, Benzo-Imidazole-H), 7.08 (d, *J* = 7.5 Hz, 2H, Benzo-imidazole-H), 3.47 (s, 2H, CH_2_), 3.11 (s, 2H, NH_2_), ^13^C NMR (150 MHz, DMSO-*d*_6_): δ 183.7, 154.5, 146.4, 138.2, 138.2, 132.3, 131.3, 131.3, 128.1, 128.1, 125.2, 122.5, 122.5, 114.7, 114.7, 36.3. HR EIMS: *m*/*z* calcd for C_16_H_15_BrN_6_S_2_ [M]^+^ 433.9160; Found: 433.9080.

#### 3.3.11. (*E*)-*N*′-(2-((1*H*-Benzo[d]imidazol-2-yl)thio)-1-(4-bromophenyl)ethylidene)hydrazinecarbothiohydrazide (**6k**)

*Light Yellowish solid*, Yield: 70%; ^1^H NMR (600 MHz, DMSO-*d*_6_): δ 12.35 (s, 1H, NH), 11.35 (s, 1H, NH), 9.43 (s, 1H, NH), 7.65 (d, *J* = 7.3 Hz, 2H, Ar-H), 7.57 (d, *J* = 7.4 Hz, 2H, Ar-H), 7.36 (d, *J* = 7.3 Hz, 2H, Benzo-Imidazole-H), 7.08 (d, *J* = 7.5 Hz, 2H, Benzo-imidazole-H), 3.47 (s, 2H, CH_2_), 3.11 (s, 2H, NH_2_), ^13^C NMR (150 MHz, DMSO-*d*_6_): δ 183.7, 154.5, 146.4, 138.2, 138.2, 132.3, 131.3, 131.3, 128.1, 128.1, 125.2, 122.5, 122.5, 114.7, 114.7, 36.3. HR EIMS: *m*/*z* calcd for C_16_H_15_BrN_6_S_2_ [M]^+^ 433.9160; Found: 433.9080.

#### 3.3.12. (*E*)-*N*′-(2-((1*H*-Benzo[d]imidazol-2-yl)thio)-1-(4-bromophenyl)ethylidene)hydrazinecarbothiohydrazide (**6l**)

*Yelowish solid*, Yield: 73%; ^1^H NMR (600 MHz, DMSO-*d*_6_): δ 12.33 (s, 1H, NH), 11.35 (s, 1H, NH), 9.43 (s, 1H, NH), 7.65 (d, *J* = 7.3 Hz, 2H, Ar-H), 7.57 (d, *J* = 7.4 Hz, 2H, Ar-H), 7.36 (d, *J* = 7.3 Hz, 2H, Benzo-Imidazole-H), 7.08 (d, *J* = 7.5 Hz, 2H, Benzo-imidazole-H), 3.47 (s, 2H, CH_2_), 3.11 (s, 2H, NH_2_), ^13^C NMR (150 MHz, DMSO-*d*_6_): δ 183.7, 154.5, 146.4, 138.2, 138.2, 132.3, 131.3, 131.3, 128.1, 128.1, 125.2, 122.5, 122.5, 114.7, 114.7, 36.3. HR EIMS: *m*/*z* calcd for C_16_H_15_BrN_6_S_2_ [M]^+^ 433.9160; Found: 433.9080.

#### 3.3.13. (*E*)-*N*′-(2-((1*H*-Benzo[d]imidazol-2-yl)thio)-1-(3-fluorophenyl)ethylidene)hydrazinecarbothiohydrazide (**6m**)

*Yelowish solid*, Yield: 75%; ^1^H NMR (600 MHz, DMSO-*d*_6_): δ 12.42 (s, 1H, NH), 11.38 (s, 1H, NH), 9.47 (s, 1H, NH), 7.77 (s, 1H, Ar-H), 7.68 (dd, *J* = 7.1, 2.3 Hz, 1H, Ar-H), 7.58 (t, *J* = 7.8 Hz, 1H, Ar-H), 7.41 (dd, *J* = 7.6, 1.9 Hz, 1H, Ar-H), 7.39 (d, *J* = 7.1 Hz, 2H, Benzo-Imidazole-H), 7.14 (d, *J* = 7.4 Hz, 2H, Benzo-imidazole-H), 3.49 (s, 2H, CH_2_), 3.18 (s, 2H, NH_2_), ^13^C NMR (150 MHz, DMSO-*d*_6_): δ 184.0, 155.5, 147.4, 139.5, 138.4, 134.3, 133.3, 132.3, 129.1, 128.1, 126.2, 124.5, 123.5, 116.7, 115.7, 37.3. HR EIMS: *m*/*z* calcd for C_16_H_15_FN_6_S_2_ [M]^+^ 374.0960; Found: 374.0840.

#### 3.3.14. (*E*)-*N*′-(2-((1*H*-Benzo[d]imidazol-2-yl)thio)-1-(3-fluorophenyl)ethylidene)hydrazinecarbothiohydrazide (**6n**)

*Light Brownish solid*, Yield: 76%; ^1^H NMR (600 MHz, DMSO-*d*_6_): δ 12.45 (s, 1H, NH), 11.38 (s, 1H, NH), 9.47 (s, 1H, NH), 7.77 (s, 1H, Ar-H), 7.68 (dd, *J* = 7.1, 2.3 Hz, 1H, Ar-H), 7.58 (t, *J* = 7.8 Hz, 1H, Ar-H), 7.41 (dd, *J* = 7.6, 1.9 Hz, 1H, Ar-H), 7.39 (d, *J* = 7.1 Hz, 2H, Benzo-Imidazole-H), 7.14 (d, *J* = 7.4 Hz, 2H, Benzo-imidazole-H), 3.49 (s, 2H, CH_2_), 3.18 (s, 2H, NH_2_), ^13^C NMR (150 MHz, DMSO-*d*_6_): δ 184.0, 155.5, 147.4, 139.5, 138.4, 134.3, 133.3, 132.3, 129.1, 128.1, 126.2, 123.5, 123.5, 115.7, 115.7, 37.3. HR EIMS: *m*/*z* calcd for C_16_H_15_FN_6_S_2_ [M]^+^ 374.0960; Found: 374.0840.

#### 3.3.15. (*E*)-*N*′-(2-((1*H*-Benzo[d]imidazol-2-yl)thio)-1-phenylethylidene)hydrazinecarbothiohydrazide (**6o**)

*Light Yelowish solid*, Yield: 77%; ^1^H NMR (600 MHz, DMSO-*d*_6_): δ 12.30 (s, 1H, NH), 11.22 (s, 1H, NH), 9.32 (s, 1H, NH), 7.80 (dd, *J* = 7.3, 1.8 Hz, 1H, Ar-H), 7.65–7.59 (m, 1H, Ar-H), 7.47 (t, *J* = 7.2 Hz, 1H, Ar-H), 7.35 (d, *J* = 7.7 Hz, 2H, Benzo-Imidazole-H), 7.07 (d, *J* = 7.3 Hz, 2H, Benzo-imidazole-H), 3.36 (s, 2H, CH_2_), 3.14 (s, 2H, NH_2_), ^13^C NMR (150 MHz, DMSO-*d*_6_): δ 183.3, 154.3, 146.2, 138.7, 138.7, 134.3, 134.3, 132.1, 128.1, 128.1, 126.8, 124.3, 123.6, 116.2, 115.1, 37.0. HR EIMS: *m*/*z* calcd for C_16_H_16_N_6_S_2_ [M]^+^ 376.7840; Found: 376.7790.

#### 3.3.16. *N*′-(2-((1*H*-Benzo[d]imidazol-2-yl)thio)-1-([1,1′-biphenyl]-4-yl)ethylidene)-2-((*E*)-2-nitrobenzylidene)hydrazine-1-carbothiohydrazide (**7**)

*Yelowish solid*, Yield: 80%; ^1^H NMR (600 MHz, DMSO-*d*_6_): *δ* 12.56 (s, 1H, NH), 9.59 (s, 1H, NH), 9.41 (s, 1H, NH), 8.97 (s, 1H, CH), 8.10 (dd, *J* = 7.1, 2.1 Hz, 1H, Ar-H), 7.95 (d, *J* = 6.9 Hz, 2H, Ar-H), 7.92 (dd, *J* = 7.8, 1.9 Hz, 1H, Ar-H), 7.91–7.87 (m, 1H, Ar-H), 7.89 (d, *J* = 7.0 Hz, 1H, Ar-H), 7.84–7.77 (m, 1H, Ar-H), 7.70 (dd, *J* = 7.5, 2.1 Hz, 2H, Ar-H), 7.69–7.65 (m, 2H, Ar-H), 7.40 (d, *J* = 7.3 Hz, 2H, Ar-H), 7.39 (t, *J* = 7.6 Hz, 2H, Ar-H), 7.10 (d, *J* = 7.0 Hz, 2H, Ar-H), 5.31 (s, 2H, CH_2_). ^13^C NMR (150 MHz, DMSO-*d*_6_): *δ* 168.0, 158.6, 148.8, 145.3, 138.6, 133.9, 133.7, 132.2, 132.1, 129.4, 129.2, 129.1, 129.1, 128.6, 127.7, 127.0, 127.0, 126.1, 124.7, 123.7, 122.2, 114.4, 113.4, 109.4, 41.2, 40.6, 40.5, 40.4, 40.0. HR EIMS: *m*/*z* calcd for C_29_H_23_N_7_O_2_S_2_ [M]^+^ 565.1241; Found: 565.1190.

#### 3.3.17. *N*′-(2-((1*H*-Benzo[d]imidazol-2-yl)thio)-1-([1,1′-biphenyl]-4-yl)ethylidene)-2-((*E*)-4-nitrobenzylidene)hydrazine-1-carbothiohydrazide (**8**)

*Honey Yelowish solid*, Yield: 83%; ^1^H NMR (600 MHz, DMSO-*d*_6_): *δ* 12.48 (s, 1H, NH), 12.50 (s, 1H, NH), 9.95 (s, 1H, NH), 8.38 (s, 1H, CH), 8.39 (d, *J* = 8.3 Hz, 2H, Ar-H), 8.28 (d, *J* = 8.5 Hz, 2H, Ar-H), 8.20 (t, *J* = 7.8 Hz, 1H, Ar-H), 8.13 (d, *J* = 8.0 Hz, 2H, Ar-H), 8.06 (d, *J* = 8.1 Hz, 2H, Ar-H), 7.95–7.83 (m, 2H, Ar-H), 7.88 (dd, *J* = 7.1, 1.9 Hz, 2H, Ar-H), 7.39 (d, *J* = 7.4 Hz, 2H, Ar-H), 7.09 (d, *J* = 7.1 Hz, 2H, Ar-H), 5.05 (s, 2H, CH_2_). ^13^C NMR (150 MHz, DMSO-*d*_6_): *δ* 168.1, 158.5, 148.7, 145.2, 138.5, 133.8, 133.6, 132.1, 132.0, 129.3, 129.1, 129.0, 129.0, 128.5, 127.6, 127.1, 127.1, 126.0, 124.6, 123.6, 122.1, 114.3, 113.3, 109.3, 41.1, 40.5, 40.4, 40.3, 40.2. HR EIMS: *m*/*z* calcd for C_29_H_23_N_7_O_2_S_2_ [M]^+^ 565.1099; Found: 565.1070.

#### 3.3.18. *N*′-(2-((1*H*-Benzo[d]imidazol-2-yl)thio)-1-([1,1′-biphenyl]-4-yl)ethylidene)-2-((*E*)-3-nitrobenzylidene)hydrazine-1-carbothiohydrazide (**9**) 

*Dijon Yelowish solid*, Yield: 87%; ^1^H NMR (600 MHz, DMSO-*d*_6_): *δ* 12.52 (s, 1H, NH), 11.51 (s, 1H, NH), 10.09 (s, 1H, OH), 8.94 (s, 1H, NH), 8.73 (s, 1H, CH), 8.46 (s, 1H, Ar-H), 8.39 (d, *J* = 8.1 Hz, 2H, Ar-H), 8.35 (d, *J* = 7.7 Hz, 2H, Ar-H), 8.05 (d, *J* = 7.8 Hz, 2H, Ar-H), 8.04 (d, *J* = 7.7 Hz, 1H, Ar-H), 8.06 (d, *J* = 8.1 Hz, 2H, Ar-H), 7.95–7.87 (m, 1H, Ar-H), 7.70 (dd, *J* = 7.2, 2.0 Hz, 2H, Ar-H), 7.40 (t, *J* = 7.1 Hz, 1H, Ar-H), 7.38 (d, *J* = 7.8 Hz, 2H, Ar-H), 7.08 (d, *J* = 7.7 Hz, 2H, Ar-H), 4.92 (s, 2H, CH_2_). ^13^C NMR (150 MHz, DMSO-*d*_6_): *δ* 191.8, 172.2, 168.0, 160.4, 148.2, 140.2, 135.2, 134.4, 132.7, 132.2, 130.6, 130.3, 130.1, 129.2, 129.1, 127.4, 127.1, 127.0, 127.0, 125.8, 124.0, 121.2, 121.1, 121.0, 122.6, 122.2, 120.6, 109.4, 20.2. HR EIMS: *m*/*z* calcd for C_29_H_23_N_7_O_2_S_2_ [M]^+^ 565.1310; Found: 565.1300.

#### 3.3.19. *N*′-(2-((1*H*-Benzo[d]imidazol-2-yl)thio)-1-([1,1′-biphenyl]-4-yl)ethylidene)-2-((*E*)-4-hydroxybenzylidene)hydrazine-1-carbothiohydrazide (**10**)

*Mustard Yelowish solid*, Yield: 77%; ^1^H NMR (600 MHz, DMSO-*d*_6_): *δ* 12.46 (s, 1H, NH), 11.05 (s, 1H, NH), 9.56 (s, 1H, OH), 8.94 (s, 1H, NH), 8.73 (s, 1H, CH), 8.17 (d, *J* = 8.2 Hz, 2H, Ar-H), 7.87 (d, *J* = 8.4 Hz,2H, Ar-H), 7.70 (dd, *J* = 7.5, 2.3 Hz, 2H, Ar-H), 7.69–7.64 (m, 2H, Ar-H), 7.63 (t, *J* = 7.1 Hz, 1H, Ar-H), 7.61 (d, *J* = 7.8 Hz, 2H, Ar-H), 7.48 (d, *J* = 9.1 Hz, 2H, Ar-H), 7.11 (d, *J* = 6.9 Hz, 2H, Ar-H), 6.81 (d, *J* = 7.8 Hz, 2H, Ar-H), 5.05 (s, 2H, CH_2_). ^13^C NMR (150 MHz, DMSO-*d*_6_): *δ* 168.3, 158.8, 148.9, 145.5, 138.7, 133.9, 133.9, 132.4, 132.3, 129.6, 129.4, 129.3, 129.3, 128.7, 127.8, 127.4, 127.4, 126.3, 124.8, 123.8, 122.4, 114.4, 113.5, 109.6, 41.4, 40.7, 40.6, 40.5, 40.3. HR EIMS: *m*/*z* calcd for C_29_H_24_N_6_OS_2_ [M]^+^ 534.1340; Found: 534.1325.

#### 3.3.20. *N*′-(2-((1*H*-Benzo[d]imidazol-2-yl)thio)-1-(3,4-dichlorophenyl)ethylidene)-2-((*E*)-2-nitrobenzylidene)hydrazine-1-carbothiohydrazide (**11**)

*Yelowish solid*, Yield: 88%; ^1^H NMR (600 MHz, DMSO-*d*_6_): *δ* 12.50 (s, 1H, NH), 10.25 (s, 1H, NH), 9.95 (s, 1H, NH), 8.97 (s, 1H, CH), 8.30 (s, 1H, Ar-H), 8.05 (dd, *J* = 7.4, 2.3 Hz, 1H, Ar-H), 7.90 (dd, *J* = 7.8, 1.9 Hz, 1H, Ar-H), 7.80 (d, *J* = 7.5 Hz, 2H, Ar-H), 7.70 (d, *J* = 6.9 Hz, 1H, Ar-H), 7.68–7.60 (m, 1H, Ar-H), 7.58–7.54 (m, 1H, Ar-H), 7.41 (d, *J* = 7.4 Hz, 2H, Ar-H), 7.14 (d, *J* = 7.6 Hz, 2H, Ar-H), 5.15 (s, 2H, CH_2_). ^13^C NMR (150 MHz, DMSO-*d*_6_): *δ* 169.3, 159.8, 149.9, 146.5, 139.7, 134.9, 134.9, 133.4, 133.3, 130.6, 130.4, 130.3, 130.3, 129.7, 128.8, 128.4, 128.4, 127.3, 124.5, 124.8, 123.4, 115.4, 114.5. HR EIMS: *m*/*z* calcd for C_23_H_17_Cl_2_N_7_O_2_S_2_ [M]^+^ 557.0150; Found: 557.0140.

#### 3.3.21. *N*′-(2-((1*H*-Benzo[d]imidazol-2-yl)thio)-1-(3,4-dichlorophenyl)ethylidene)-2-((*E*)-4-hydroxybenzylidene)hydrazine-1-carbothiohydrazide (**12**)

*Yelowish solid*, Yield: 79%; ^1^H NMR (600 MHz, DMSO-*d*_6_): *δ* 12.39 (s, 1H, NH), 10.59 (s, 1H, NH), 9.85 (s, 1H, NH), 9.71 (s, 1H, OH), 8.63 (s, 1H, CH), 8.10 (s, 1H, Ar-H), 7.87 (d, *J* = 6.9 Hz, 1H, Ar-H), 7.70 (d, *J* = 7.3 Hz, 1H, Ar-H), 7.58 (d, *J* = 7.8 Hz, 2H, Ar-H), 7.47 (d, *J* = 7.6 Hz, 2H, Ar-H), 7.10 (d, *J* = 6.8 Hz, 2H, Ar-H), 6.88 (d, *J* = 7.9 Hz, 2H, Ar-H), 5.20 (s, 2H, CH_2_). ^13^C NMR (150 MHz, DMSO-*d*_6_): *δ* 169.3, 159.4, 149.5, 146.5, 139.8, 134.5, 134.3, 133.6, 133.5, 130.6, 130.4, 130.3, 130.2, 129.9, 128.6, 128.6, 128.5, 127.4, 124.5, 124.4, 123.7, 115.8, 114.9. HR EI-MS: *m*/*z* calcd for C_23_H_18_Cl_2_N_6_OS_2_ [M]^+^ 528.0248; Found: 528.0237.

#### 3.3.22. *N*′-(2-((1*H*-Benzo[d]imidazol-2-yl)thio)-1-(3,4-dichlorophenyl)ethylidene)-2-((*E*)-4-nitrobenzylidene)hydrazine-1-carbothiohydrazide (**13**)

*Yelowish solid*, Yield: 84%; ^1^H NMR (600 MHz, DMSO-*d*_6_): *δ* 12.47 (s, 1H, NH), 10.17 (s, 1H, NH), 9.95 (s, 1H, NH), 8.88 (s, 1H, CH), 8.43 (d, *J* = 8.1 Hz, 2H, Ar-H), 8.18 (d, *J* = 8.1 Hz, 2H, Ar-H), 8.02 (d, *J* = 8.2 Hz, 1H, Ar-H), 7.60 (d, *J* = 7.1 Hz, 1H, Ar-H), 7.55 (s, 1H, Ar-H), 7.38 (d, *J* = 7.6 Hz, 2H, Ar-H), 7.15 (d, *J* = 8.3 Hz, 2H, Ar-H), 5.15 (s, 2H, CH_2_). ^13^C NMR (150 MHz, DMSO-*d*_6_): *δ* 169.1, 159.6, 149.7, 146.3, 139.5, 134.7, 134.7, 133.2, 133.1, 130.4, 130.2, 130.1, 130.1, 129.7, 128.6, 128.2, 128.2, 127.1, 124.3, 124.6, 123.2, 115.2, 114.3. HR EIMS: *m*/*z* calcd for C_23_H_17_Cl_2_N_7_O_2_S_2_ [M]^+^ 557.0140; Found: 557.0130.

#### 3.3.23. *N*′-(2-((1*H*-Benzo[d]imidazol-2-yl)thio)-1-(3,4-dichlorophenyl)ethylidene)-2-((*E*)-3-nitrobenzylidene)hydrazine-1-carbothiohydrazide (**14**)

*Yelowish solid*, Yield: 81%; ^1^H NMR (600 MHz, DMSO-*d*_6_): *δ* 12.51 (s, 1H, NH), 11.51 (s, 1H, NH), 10.15 (s, 1H, NH), 8.94 (s, 1H, CH), 8.73 (s, 1H, Ar-H), 8.05 (d, *J* = 7.3 Hz, 2H, Ar-H), 7.82 (d, *J* = 6.9 Hz, 1H, Ar-H), 7.79 (s, 1H, Ar-H), 7.69 (d, *J* = 6.9 Hz, 1H, Ar-H), 7.63 (d, *J* = 6.0 Hz, 2H, Ar-H), 7.39 (d, *J* = 6.1 Hz, 2H, Ar-H), 7.15 (d, *J* = 9.1 Hz, 2H, Ar-H), 3.73 (s, 2H, CH_2_). ^13^C NMR (150 MHz, DMSO-*d*_6_): *δ* 191.7, 168.1, 160.4, 148.2, 137.1, 135.2, 135.1, 134.8, 134.4, 132.2, 131.9, 131.2, 130.9, 130.6, 130.4, 128.5, 128.4, 125.8, 124.0, 122.6, 122.2, 114.4, 109.2. HR EIMS: *m*/*z* calcd for C_23_H_17_Cl_2_N_7_O_2_S_2_ [M]^+^ 557.0140; Found: 557.0130.

#### 3.3.24. *N*′-(2-((1*H*-Benzo[d]imidazol-2-yl)thio)-1-(4-bromophenyl)ethylidene)-2-((*E*)-2-nitrobenzylidene)hydrazine-1-carbothiohydrazide (**15**)

*Yelowish solid*, Yield: 80%; ^1^H NMR (600 MHz, DMSO-*d*_6_): *δ* 12.59 (s, 1H, NH), 10.25 (s, 1H, NH), 9.49 (s, 1H, NH), 8.97 (s, 1H, CH), 8.11 (dd, *J* = 7.9, 1.9 Hz, 1H, Ar-H), 8.00 (dd, *J* = 7.8, 2.1 Hz, 1H, Ar-H), 7.90–7.77 (m, 1H, Ar-H), 7.67–7.61 (m, 1H, Ar-H), 7.60 (d, *J* = 7.2 Hz, 2H, Ar-H), 7.18 (d, *J* = 6.9 Hz, 2H, Ar-H), 7.46 (d, *J* = 7.6 Hz, 2H, Ar-H), 7.07 (d, *J* = 7.8 Hz, 2H, Ar-H), 3.73 (s, 2H, CH_2_). ^13^C NMR (150 MHz, DMSO-*d*_6_): *δ* 191.6, 168.0, 160.3, 148.1, 137.0, 135.1, 135.0, 134.7, 134.3, 132.1, 131.8, 131.1, 130.8, 130.5, 130.3, 128.4, 128.3, 125.7, 124.0, 122.5, 122.1, 114.3, 109.1. HR EIMS: *m*/*z* calcd for C_23_H_18_BrN_7_O_2_S_2_ [M]^+^ 567.0045; Found: 567.0030.

#### 3.3.25. *N*′-(2-((1*H*-Benzo[d]imidazol-2-yl)thio)-1-(4-bromophenyl)ethylidene)-2-((*E*)-4-hydroxybenzylidene)hydrazine-1-carbothiohydrazide (16)

*Yelowish solid*, Yield: 74%; ^1^H NMR (600 MHz, DMSO-*d*_6_): *δ* 12.42 (s, 1H, NH), 10.20 (s, 1H, NH), 9.54 (s, 1H, NH), 8.99 (s, 1H, CH), 8.15 (d, *J* = 7.8 Hz, 2H, Ar-H), 8.07 (d, *J* = 7.6 Hz, 2H, Ar-H), 7.60 (d, *J* = 7.5 Hz, 2H, Ar-H), 7.59 (d, *J* = 7.1 Hz, 2H, Ar-H), 7.18 (d, *J* = 6.9 Hz, 2H, Ar-H), 6.80 (d, *J* = 7.1 Hz, 2H, Ar-H), 2.87 (s, 2H, CH_2_). ^13^C NMR (150 MHz, DMSO-*d*_6_): *δ* 191.6, 168.0, 160.3, 148.1, 137.0, 135.1, 135.0, 134.7, 134.3, 132.1, 131.8, 131.1, 130.8, 130.5, 130.3, 128.4, 128.3, 125.7, 124.0, 122.5, 122.1, 114.3, 109.1. HR EIMS: *m*/*z* calcd for C_23_H_19_BrN_6_OS_2_ [M]^+^ 538.0140; Found: 538.0132.

#### 3.3.26. *N*′-(2-((1*H*-Benzo[d]imidazol-2-yl)thio)-1-(4-bromophenyl)ethylidene)-2-((*E*)-4-nitrobenzylidene)hydrazine-1-carbothiohydrazide (**17**)

*Yelowish solid*, Yield: 85%; ^1^H NMR (600 MHz, DMSO-*d*_6_): *δ* 12.48 (s, 1H, NH), 10.17 (s, 1H, NH), 9.57 (s, 1H, NH), 8.88 (s, 1H, CH), 8.43 (d, *J* = 8.3 Hz, 2H, Ar-H), 8.39 (d, *J* = 8.5 Hz, 2H, Ar-H), 8.17 (d, *J* = 8.3 Hz, 2H, Ar-H), 8.01 (d, *J* = 8.2 Hz, 2H, Ar-H), 7.41 (d, *J* = 7.5 Hz, 2H, Ar-H), 7.15 (d, *J* = 7.4 Hz, 2H, Ar-H), 3.60 (s, 2H, CH_2_). ^13^C NMR (150 MHz, DMSO-*d*_6_): *δ* 191.3, 167.0, 159.3, 147.1, 136.0, 134.1, 134.0, 133.7, 133.3, 131.1, 130.8, 130.1, 129.8, 129.5, 129.3, 127.4, 127.3, 124.7, 123.0, 121.5, 121.1, 113.3, 108.1. HR EIMS: *m*/*z* calcd for C_23_H_18_BrN_7_O_2_S_2_ [M]^+^ 567.0040; Found: 567.0030.

#### 3.3.27. *N*′-(2-((1*H*-Benzo[d]imidazol-2-yl)thio)-1-(4-bromophenyl)ethylidene)-2-((*E*)-3-nitrobenzylidene)hydrazine-1-carbothiohydrazide (**18**)

*Yelowish solid*, Yield: 87%; ^1^H NMR (600 MHz, DMSO-*d*_6_): *δ* 12.62 (s, 1H, NH), 10.15 (s, 1H, NH), 9.59 (s, 1H, NH), 8.94 (s, 1H, CH), 8.73 (s, 1H, Ar-H), 8.40 (d, *J* = 8.1 Hz, 2H, Ar-H), 8.35 (d, *J* = 7.6 Hz, 2H, Ar-H), 8.01 (d, *J* = 8.1 Hz, 2H, Ar-H), 8.00 (d, *J* = 7.8 Hz, 2H, Ar-H), 7.45 (d, *J* = 7.9 Hz, 2H, Ar-H), 7.14 (d, *J* = 8.2 Hz, 2H, Ar-H), 5.23 (s, 2H, CH_2_). ^13^C NMR (150 MHz, DMSO-*d*6): *δ* 191.2, 167.1, 159.2, 147.0, 136.1, 134.0, 134.1, 133.8, 133.4, 131.2, 130.7, 130.2, 129.7, 129.4, 129.2, 127.3, 127.2, 124.6, 123.1, 121.4, 121.2, 113.4, 108.2. HR EIMS: *m*/*z* calcd for C_23_H_18_BrN_7_O_2_S_2_ [M]^+^ 567.0040; Found: 567.0010.

#### 3.3.28. *N*′-((*Z*)-2-((1*H*-Benzo[d]imidazol-2-yl)thio)-1-(3-fluorophenyl)ethylidene)-2-((*Z*)-2-fluorobenzylidene)hydrazine-1-carbothiohydrazide (**19**)

*Yelowish solid*, Yield: 82%; ^1^H NMR (600 MHz, DMSO-*d*_6_): *δ* 12.49 (s, 1H, NH), 10.13 (s, 1H, NH), 9.77 (s, 1H, NH), 7.86 (d, *J* = 7.7 Hz, 2H, Ar-H), 7.78 (d, *J* = 7.2 Hz, 2H, Ar-H), 7.80 (s, 1H, C-H), 7.59 (d, *J* = 7.1 Hz, 1H, Thiazole-H), 7.49 (d, *J* = 6.8 Hz, 1H, Thiazole-H). ^13^C NMR (150 MHz, DMSO-*d*_6_): *δ* 173.8, 171.7, 169.2, 167.6, 145.4, 131.6, 130.7, 130.7, 130.3, 130.9, 130.9, 129.2, 128.7. HR EIMS: *m*/*z* calcd for C_13_H_8_N_4_O_3_S_2_ [M]^+^ 332.0037; Found: 332.0106. 

#### 3.3.29. *N*′-((*Z*)-2-((1*H*-Benzo[d]imidazol-2-yl)thio)-1-(3-fluorophenyl)ethylidene)-2-((*E*)-2-methyl-4-nitrobenzylidene)hydrazine-1-carbothiohydrazide (**20**) 

*Brownish Yelowish solid*, Yield: 76%; ^1^H NMR (600 MHz, DMSO-*d*_6_): *δ* 12.57 (s, 1H, NH), 11.70 (s, 1H, NH), 9.53 (s, 1H, NH), 10.17 (s, 1H, CH), 8.39 (d, *J* = 8.4 Hz, 1H, Ar-H), 8.18 (d, *J* = 8.2 Hz, 1H, Ar-H), 8.08 (s, 1H, Ar-H), 7.95 (t, *J* = 8.1 Hz, 1H, Ar-H), 7.85 (dd, *J* = 6.0, 3.1 Hz, 1H, Ar-H), 7.57 (dd, *J* = 6.1, 3.1 Hz, 1H, Ar-H), 7.40 (d, *J* = 7.9 Hz, 2H, Ar-H), 7.11 (d, *J* = 7.3 Hz, 2H, Ar-H), 2.24 (s, 2H, CH_2_), 1.99 (s, 3H, CH_3_). ^13^C NMR (150 MHz, DMSO-*d*_6_): *δ* 192.4, 168.5, 160.6, 148.4, 137.7, 135.8, 135.6, 134.8, 134.4, 133.2, 132.7, 132.2, 131.7, 131.4, 131.2, 129.3, 129.2, 126.6, 125.1, 122.4, 122.2, 114.4, 109.2. HR EIMS: *m*/*z* calcd for C_24_H_20_FN_7_O_2_S_2_ [M]^+^ 521.1001; Found: 521.0982.

#### 3.3.30. *N*′-((*Z*)-2-((1*H*-Benzo[d]imidazol-2-yl)thio)-1-phenylethylidene)-2-((*E*)-3-nitrobenzylidene)hydrazine-1-carbothiohydrazide (**21**)

*Yelowish solid*, Yield: 86%; ^1^H NMR (600 MHz, DMSO-*d*_6_): *δ* 12.37 (s, 1H, NH), 11.50 (s, 1H, NH), 9.68 (s, 1H, NH), 8.94 (s, 1H, CH), 8.73 (s, 1H, Ar-H), 8.01 (d, *J* = 7.6 Hz, 2H, Ar-H), 7.90 (dd, *J* = 7.8 Hz, 2H, Ar-H), 7.85 (d, *J* = 7.7 Hz, 1H, Ar-H), 7.84 (dd, *J* = 7.9 Hz, 1H, Ar-H), 7.53–7.45 (m, 2H, Ar-H), 7.39 (d, *J* = 7.6 Hz, 2H, Ar-H), 7.16 (d, *J* = 8.1 Hz, 2H, Ar-H), 2.24 (s, 2H, CH_2_). ^13^C NMR (150 MHz, DMSO-*d*_6_): *δ* 162.4, 167.5, 161.6, 149.4, 138.7, 134.8, 133.6, 133.8, 133.4, 132.2, 131.7, 131.2, 130.7, 130.4, 130.2, 128.3, 128.2, 125.6, 124.1, 122.4, 122.2, 114.4, 109.2. HR EIMS: *m*/*z* calcd for C_23_H_19_N_7_O_2_S_2_ [M]^+^ 489.1020; Found: 489.1006.

### 3.4. Molecular Docking

A molecular docking study was conducted by using the discovery studio visualizer (DSV) MGL tool 1.5.7 and autodock vina [32,33,34]. In this study, the synthesized moieties were analyzed against α-Glucosidase and α-amylase enzymes. The structures of these enzymes were retrieved from the protein data bank (PDB) with searching codes 1b2y & 3w37. In the first step, the protein was prepared by using DSV, in which water molecules and already present ligands were removed, and the target protein and prepared ligand were saved in PDB format. It was further carried out in an autodock in which polar hydrogen and Kollman and gasteiger charges were added. The selected ligand was prepared by using a torsion tree to detect the root. Moreover, the configuration file was generated along with the X, Y, and Z axis, saving both the ligand and protein in PDBQT format in the same docking folder. At the end command prompt was used to generate varied poses of the ligand; thus, nine different poses were obtained in the PDBQT format, as shown in Table 2 and Table 3. These selected analogs showed better interactions against both α-amylase and α-glucosidase. The dock protein and ligand were then opened in DSV to identify the binding residue with the active sites of the ligand.

### 3.5. α-Amylase Activity Assay

α-Amylase inhibition was determined by an assay modified by Kwon, Apostolidis & Shetty [35,36]. A total of 40 μL of sample and 40 μL of 0.02 M sodium phosphate buffer (pH 6.9 with 0.006 M sodium chloride) containing α-amylase solution (Porcine pancreatic α-amylase) (0.5 mg/mL) were incubated at 25 °C for 10 min. After pre-incubation, 40 μL of a 1% starch solution in 0.02 M sodium phosphate buffer (pH 6.9 with 0.006 M sodium chloride) was added to each tube at 5 s intervals. The reaction mixtures were then incubated at 25 °C for 10 min. The reaction was stopped with 100 μL of dinitrosalicylic acid color reagent. The test tubes were then incubated in a boiling water bath for 5 min and cooled to room temperature. The reaction mixture was then diluted after adding 900 μL of distilled water, and the absorbance was measured at 540 nm. Acarbose was used as a reference drug. All reactions were carried out in triplicate. Calculation of the concentration of compound required to scavenge 50% of the radical (IC_50_) as per the formula below:I% = (Ac − As)/Ac × 100
Ac = the absorbance of the control
As = the absorbance of the sample.

### 3.6. α-Glucosidase Activity Assay

The α-Glucosidase activity was determined by a slight modification to the already-reported method [37]. In a 96-well microplate, 10 μL of test samples (5 mg/mL DMSO solution) were reconstituted in 100 μL of 100 mM-phosphate buffer (pH 6.8) and incubated with 50 μL of crude intestinal α-glucosidase for 5 min before 50 μL substrate (5 mM, p-nitrophenyl-α-D-glucopyranoside prepared in the same buffer) was added. The release of p-nitrophenol was measured at 405 nm spectrophotometrically for 5 min after incubation with the substrate. Individual blanks for test samples were prepared to correct background absorbance where the substrate was replaced with 50 μL of the buffer. The control sample contained 10 μL DMSO in place of the test samples. All reactions were carried out in triplicate.

The percentage of enzyme inhibition was calculated as (1 − B/A) × 100.

Where A represents the absorbance of control without test samples, and B represents absorbance in the presence of test samples.

## Data Availability

Not applicable.

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
