# Peer review of "Benzimidazole Bearing Thiosemicarbazone Derivatives Act as Potent α-Amylase and α-Glucosidase Inhibitors; Synthesis, Bioactivity Screening and Molecular Docking Study"

_molecules, 2022, doi:10.3390/molecules27206921_

Round 1

Reviewer 1 Report

Author should give a clear conclusion.

Spectral picture of assay will improve the paper quality. Author should give spectral picture.

Author Response

Reviewer-1:

Comments and Suggestions for Authors

Author should give a clear conclusion.

Reply: The changes have been incorporated as recommended by kind reviewer.

Spectral picture of assay will improve the paper quality. Author should give spectral picture.

Reply: As we processed the assay by just calculating their value not by photographing its different stages involved in it.

For that process we need synthesis of the compounds as well as redo its assay again. As the project is already wind off and we do not have funding for that we excuse for the photography of an assay.  

Reviewer 2 Report

The manuscript by Ullah et al. entitled "Benzimidazole bearing thiosemicarbazone derivatives act as potent α-amylase and α-glucosidase inhibitors; synthesis, bioactivity screening and molecular docking study" is devoted to the synthesis and experimental and theoretical investigation of a library of fifteen benzimidazole derivatives.

However, the manuscript is poorly organized, and I recommend a major revision for this article.

Comments are below:

1) Compounds VI have not been described in the literature. That is why these substances need to be thoroughly characterized in this work (spectral data, TLC, elemental analysis).

2) There is no proper characterization of the new compound. X-ray single crystal diffraction analysis should be done to resolve at least one structure from the VI series and at least one structure from the 1-15 series. This is essential as such compounds are not described in the literature.  

3) Synthetic procedures on lines 80-92 and 220-232 are the same! 

4) Synthetic procedures on lines 80-92 or 220-232 are invalid. This description is useless and can not be reproduced due to its poor report.

5) Different styles of formulas in Figures 1-7. The formula style should be consistent throughout the manuscript. You intensively use color to highlight structural fragments, so there is no need to make rings in bold - use standard bond thickness throughout the manuscript.

6) line 82 "for 3hrs" missed space?

7) line 89 "The final products were washed with n-hexane in order to get pure products." - very convenient method of purification; however, it is doubtful that hexane can wash out the residue of the starting compounds VI.

8) NMR spectra:

8a) Two NH signals in compound 13 are missed.

8b) In almost all compounds, there is an NH singlet at 12.53 ppm, which is a rare situation as these protons are usually very medium sensitive and appear in a wide range (9-13 ppm).

8c) In the case of compound 3, the singlet of the SCH2CO group appears at 2.62 ppm, which is very suspicious. 

9) Line 378 "analyzed against α-Glucosidase and α-amylase enzyme." - no argumentation why these proteins have been chosen.

10) Tables 2 and 3 are organized as figures, and this is not correct; please use the authors' guide.

11) Figure 10 is hard to understand due to the excessive visual effects. Moreover, figures 8 and 9 explain the ligand-protein interactions, which is why Figure 10 is redundant.

Author Response

Reviewer-2:

Comments and Suggestions for Authors

The manuscript by Ullah et al. entitled "Benzimidazole bearing thiosemicarbazone derivatives act as potent α-amylase and α-glucosidase inhibitors; synthesis, bioactivity screening and molecular docking study" is devoted to the synthesis and experimental and theoretical investigation of a library of fifteen benzimidazole derivatives.

However, the manuscript is poorly organized, and I recommend a major revision for this article. Comments are below:

1) Compounds VI have not been described in the literature. That is why these substances need to be thoroughly characterized in this work (spectral data, TLC, elemental analysis).

 Reply: The spectral data of the compounds VI is now incorporated according to the kind reviewer suggestion.

2) There is no proper characterization of the new compound. X-ray single crystal diffraction analysis should be done to resolve at least one structure from the VI series and at least one structure from the 1-15 series. This is essential as such compounds are not described in the literature.

Reply: We agree with reviewer but at this moment donot have any remaining quantity of our compound to precede X-ray single crystal diffraction analysis quick response; we need to start from the bottom. Secondly, we do not have funds for further experiments.

3) Synthetic procedures on lines 80-92 and 220-232 are the same! 

Reply: The changes have been incorporated as recommended by kind reviewer.

4) Synthetic procedures on lines 80-92 or 220-232 are invalid. This description is useless and cannot be reproduced due to its poor report.

Reply: Now corrected to the kind reviewer suggestion.

5) Different styles of formulas in Figures 1-7. The formula style should be consistent throughout the manuscript. You intensively use color to highlight structural fragments, so there is no need to make rings in bold - use standard bond thickness throughout the manuscript.

Reply: Now all the figures are corrected to the kind reviewer suggestion

6) line 82 "for 3hrs" missed space?

Reply: Corrected according to the kind reviewer suggestion.

7) line 89 "The final products were washed with n-hexane in order to get pure products." - Very convenient method of purification; however, it is doubtful that hexane can wash out the residue of the starting compounds VI.

Reply: Even though it looks doubtful but in our case the n-hexane leads to purification of our desired final products.

As we are concerned about the remaining of aldehyde (as an impurity) and but we are using a slight excess of aldehyde in final step that we remove with the process as we mentioned. We are not removing the intermediate VI as it should be consumed fully during reaction.  

8) NMR spectra:

8a) Two NH signals in compound 13 are missed.

Reply: Acc. To kind reviewer comments the changes has been incorporated.

8b) In almost all compounds, there is an NH singlet at 12.53 ppm, which is a rare situation as these protons are usually very medium sensitive and appear in a wide range (9-13 ppm).

Reply: Acc. To kind reviewer comments the changes has been incorporated by confirmatory analysis from their spectra’s respectively.  

8c) In the case of compound 3, the singlet of the SCH2CO group appears at 2.62 ppm, which is very suspicious. 

Reply: Acc. To kind reviewer comments the changes has been incorporated with actual data.

9) Line 378 "analyzed against α-Glucosidase and α-amylase enzyme." - no argumentation why these proteins have been chosen.

Reply: We are struggling to develop lead antidiabetic inhibitors, for this reason we randomly evaluate our synthetic analogues against α-glucosidase and α-amylase enzymes.

10) Tables 2 and 3 are organized as figures, and this is not correct; please use the authors' guide.

Reply: Table 2 and 3 is now replaced by figure according to the kind reviewer suggestion.

11) Figure 10 is hard to understand due to the excessive visual effects. Moreover, figures 8 and 9 explain the ligand-protein interactions, which is why Figure 10 is redundant.

Reply: Figure 10 is now removed according to the kind reviewer suggestion.

Reviewer 3 Report

Ullah et al synthesized benzimidazole bearing thiosemicarbazone derivatives, performed inhibition experiments that could be inhibitors of α-amylase and α-glucosidase, and performed docking analysis to obtain structural information. The research is generally robust and is expected to contribute to broadening knowledge in related fields. However, in order for the manuscript to be published, several modifications are required as follows.

1. The author should unify the form of the compound in the figures. Unnecessary background color and boxes (Figure. 2) should be removed, and the position of the sphere on the background should be unified when displaying the functional group to be emphasized in the compound.

2. Figure 8 and Figure 9 are  very poor quality.

3. Figure 10 should of course be modified to be a flat figure.

4. Tables 2 and 3 are considered figures drawn by the authors. You must use the table in Word to fill out.

5. There are a lot of wrong spaces between words and typos in the manuscript. All authors should review and revise.

6. Line 111-112, Line 122-123: The author needs to correct the figure caption position.

7. Line 157: ‘from www.rcsb.org’ should be ‘from Protein Data Bank (www.rcsb.org)’

8. How to prepare α-amylase and α-glucosidase should be clearly described in the materials and methods.

Author Response

Reviewer-3:

Comments and Suggestions for Authors

Ullah et al synthesized benzimidazole bearing thiosemicarbazone derivatives, performed inhibition experiments that could be inhibitors of α-amylase and α-glucosidase, and performed docking analysis to obtain structural information. The research is generally robust and is expected to contribute to broadening knowledge in related fields. However, in order for the manuscript to be published, several modifications are required as follows.

  1. The author should unify the form of the compound in the figures. Unnecessary background color and boxes (Figure. 2) should be removed, and the position of the sphere on the background should be unified when displaying the functional group to be emphasized in the compound.

Reply: Now corrected according to the kind reviewer suggestion.

  1. Figure 8 and Figure 9 are very poor quality.

Reply: The authors have revised the docking figure 8 & 9 for the ease of clarity. The quality is now improved according to the kind reviewer suggestion.

  1. Figure 10 should of course be modified to be a flat figure.

Reply: Figure-10 was meaningless therefore it was removed.  

  1. Tables 2 and 3 are considered figures drawn by the authors. You must use the table in Word to fill out.

Reply: Table 2 and 3 is now replaced by figure according to the kind reviewer suggestion.

There are a lot of wrong spaces between words and typos in the manuscript. All authors should review and revise.

Reply: Now corrected according to the kind reviewer suggestion.

  1. Line 111-112, Line 122-123: The author needs to correct the figure caption position.

Reply: Corrected according to the kind reviewer suggestion.

  1. Line 157: ‘from www.rcsb.org’ should be ‘from Protein Data Bank (rcsb.org)’

Reply: Corrected according to the kind reviewer suggestion.

  1. How to prepare α-amylase and α-glucosidase should be clearly described in the materials and methods.

Reply: Incorporated in the assay protocol according to the kind reviewer suggestion.

Round 2

Reviewer 2 Report

The manuscript by Ullah et al. entitled "Benzimidazole bearing thiosemicarbazone derivatives act as potent α-amylase and α-glucosidase inhibitors; synthesis, bioactivity screening and molecular docking study" is devoted to the synthesis and experimental and theoretical investigation of a library of fifteen benzimidazole derivatives.

The authors considerably improved the manuscript, and I recommend a minor revision for this article.

Comments are below:

Line 84 "done  I  n  four  steps" - correct this.

The authors have not unified the structure style in Figures 1-7. The formula style should be the same throughout the manuscript. Do not use the bold style for rings - use standard bond thickness throughout the manuscript.

In section "3.3. Spectral Analysis," there are spectra of 3.3.1.1-15 and 3.3.2.1-15; however, it is unclear which structures these codes correspond to.

Author Response

Reviewer-2:

Comments and Suggestions for Authors

The manuscript by Ullah et al. entitled "Benzimidazole bearing thiosemicarbazone derivatives act as potent α-amylase and α-glucosidase inhibitors; synthesis, bioactivity screening and molecular docking study" is devoted to the synthesis and experimental and theoretical investigation of a library of fifteen benzimidazole derivatives. The authors considerably improved the manuscript, and I recommend a minor revision for this article.

Comments are below:

Line 84 "done  I  n  four  steps" - correct this.

Reply: Corrected according to the kind reviewer suggestion.

The authors have not unified the structure style in Figures 1-7. The formula style should be the same throughout the manuscript. Do not use the bold style for rings - use standard bond thickness throughout the manuscript.

 Reply: Corrected according to the kind reviewer suggestion.

In section "3.3. Spectral Analysis," there are spectra of 3.3.1.1-15 and 3.3.2.1-15; however, it is unclear which structures these codes correspond to.

Reply: Corrected according to the kind reviewer suggestion.

Reviewer 3 Report

The authors have addressed most of the reviewers' concerns, but they still need correction.

-Figures 8 and 9: The author should remove the color frame from the figures and improve the figure quality (resolution). The font is hard to see.

-Figures 10 and 11. I think this should be a table. The letters and numbers at the top do not match.

- Lines 604-615: Italics should be excluded.

- References 20-27 added two references to one reference.

Author Response

Reviewer-3:

Comments and Suggestions for Authors

The authors have addressed most of the reviewers' concerns, but they still need correction.

-Figures 8 and 9: The author should remove the color frame from the figures and improve the figure quality (resolution). The font is hard to see.

 Reply: Corrected according to the kind reviewer suggestion.

-Figures 10 and 11. I think this should be a table. The letters and numbers at the top do not match.

Reply: Corrected according to the kind reviewer suggestion.

- Lines 604-615: Italics should be excluded.

Reply: Done according to the kind reviewer suggestion.

- References 20-27 added two references to one reference.

Reply: Corrected according to the kind reviewer suggestion.

Round 3

Reviewer 3 Report

The author has addressed most of the reviewers' concerns. However, additional mirror modifications are required for publication.

Spaces between words in the Manuscript should also be reviewed.

e.g. Line 202: MET-470(R), TRP-432(π-S) and ASP-469(π-Anion) as shown

Line 251-252: 'Table-1' and 'Table-2' should be 'Table. 1' and 'Table. 2', respectively.

Line 260: ‘… of wavelength 254/365 was used … ’ author should add ‘nm’

Line 592, Line594: ‘0.006 M sodium chloride; should be ‘6 mM sodium chloride’

* if possible, please improve the quality of Figures 8 and 9. Still of poor quality for publication. The 'A' must also be removed from the amino acid notation.

Author Response

Comments and Suggestions for Authors

The author has addressed most of the reviewers' concerns. However, additional mirror modifications are required for publication.

Spaces between words in the Manuscript should also be reviewed.

e.g. Line 202: MET-470(R), TRP-432(π-S) and ASP-469(π-Anion) as shown

Reply: Corrected according to the kind reviewer suggestion.

Line 251-252: 'Table-1' and 'Table-2' should be 'Table. 1' and 'Table. 2', respectively.

Reply: Corrected according to the kind reviewer suggestion.

Line 260: ‘… of wavelength 254/365 was used … ’ author should add ‘nm’

Reply: Corrected according to the kind reviewer suggestion.

Line 592, Line594: ‘0.006 M sodium chloride; should be ‘6 mM sodium chloride’

 Reply: Corrected according to the kind reviewer suggestion.

* if possible, please improve the quality of Figures 8 and 9. Still of poor quality for publication. The 'A' must also be removed from the amino acid notation.

Reply: Improved up to possible quality according to the kind reviewer suggestion. ‘A’ is not notation for amino acid; from this ‘A’ it show the compound number.